# Evaluation of a patient and public involvement training programme for researchers at a large biomedical research centre in the UK

Rosamund Yu  ,[1] Bec Hanley,[2] Simon Denegri,[3] Jaber Ahmed,[1] Nicholas J McNally[1]

[1]NIHR Biomedical Research Centre at UCLH, University College London Hospitals NHS Foundation Trust, London, UK
[2]Medical Research Council Clinical Trials Unit at University College London, London, UK
[3]Academy of Medical Sciences, London, UK

**Correspondence to**
Dr Rosamund Yu;
rosamund.yu@ucl.ac.uk

## ABSTRACT

**Objectives** To design, deliver and evaluate a programme of training workshops for biomedical researchers aimed at building confidence and skills in actively involving patients and the public (PPI) in research.

**Design** A bespoke programme of training workshops in PPI aimed at researchers.

**Setting** A large National Institute for Health Research Biomedical Research Centre in London and several partner organisations.

**Participants** 721 scientists, clinicians and research managers attending dedicated training in PPI at a major London NHS (National Health Service)–university partnership.

**Interventions** A programme of 72 training workshops, designed to build practical skills and confidence for researchers working with patients and the public in research, was delivered at a major research-active NHS:university partnership. An iterative approach was taken to the programme, with the content of the workshops continually reviewed and refreshed to respond to the needs of researchers. Surveys before, immediately following and 6 months after training investigated the impact on researchers' confidence and skills in PPI work, and the kind of PPI they subsequently carried out.

**Results** Training brought about immediate marked increases in researchers' self-reported confidence to carry out PPI activities within their research, and in their knowledge of good practice. The evaluation indicates that workshop attendees were more likely to involve patients in their research following training. Researchers tended to involve patients and the public in a range of areas, including input to study design and patient information, in particular.

**Conclusions** When positioned within a broader organisational strategy for PPI in research, such training has an important role to play in progressing PPI in a major research partnership. Training appeared to provide the confidence needed to carry out PPI which enabled further development of confidence and skills. Involving researchers who have attended the training in the ongoing development of the programme and bringing in patients to the training programme are key next steps.

## INTRODUCTION

In the UK, patient and public involvement (PPI) in biomedical research has been encouraged and promoted over the last 15 years. Evidence of PPI has become a condition of many research funders, notably the National Institute for Health Research (NIHR), which has helped to set expectations for PPI.[1] The NIHR's standards for public involvement are designed to improve the quality and consistency of PPI in health research, emphasising the importance of inclusive opportunities, working together, support and learning, communications, impact and governance, as values-based areas for reflection and learning for researchers and research organisations.[2]

There are two dominant narratives underpinning the importance of PPI in biomedical research. First is the notion that PPI improves the quality, relevance and acceptability of research.[3–5] The second is that PPI enables citizens to exercise their rights. This can mean that

they feel empowered and valued and that they can build their skills and/or knowledge.[6 7] The power dynamic that exists between healthcare professionals, researchers and patients can be deep rooted.[8–10] The primacy that is given to clinical or scientific knowledge over the experiential knowledge that patients bring to the research process has been shown to render much involvement practice as tokenistic.[11] There are considerable challenges of establishing meaningful PPI in hierarchical, scientific research organisational settings.[12]

## Strengths and limitations of this study

► This study evaluates probably one of the largest patient and public involvement (PPI) in biomedical research training programmes in the UK, carried out across a major research partnership.
► The precise content of the training workshops was iterative and evolved throughout the programme in response to feedback from participants, making it highly tailored to researchers' needs.
► The evaluation was carefully designed to monitor the impact of PPI training on researchers and their work. However, it has not sought the perspectives of patients participating in projects being carried out by the researchers.
► Response rates to the surveys carried out immediately before and after training were high (average 98%). However, response rates to the survey carried out at 6 months post-training were low (34%) and respondents were self-selecting and their responses were dependent on researchers' self-evaluation and recall.

Researchers' experiences and attitudes towards PPI are undoubtedly key to embedding involvement within the wider research culture of a research organisation.[13] Dedicated training in PPI has long been identified as a need[14] and an important mechanism for developing researchers' skills, experiences and attitudes to PPI.[15–19] Researchers are also being challenged to document in a structured way how they involve patients throughout the research process to ensure PPI practice is based on the best evidence.[20 21]

In England, the National Institute for Health Research Biomedical Research Centres (BRCs) are partnerships between National Health Service (NHS) Trusts and universities. The BRCs are funded by the NIHR to drive experimental medicine research, taking promising scientific concepts from laboratories into early-stage studies in patient populations, for the purposes of establishing the evidence base that will enable new therapies and diagnostics to progress to clinical practice and patient benefit at pace.[22] One condition of NIHR BRC funding is that the BRCs must have robust strategic plans in place for PPI. The NIHR University College London Hospitals BRC, a partnership between University College London Hospitals (UCLH) and University College London (UCL), is one of the largest BRCs. The UCLH/UCL partnership has a portfolio of over 1000 clinical research studies, opening 300 new studies every year and over 600 principal investigators.

Overseeing the UCLH BRC's strategy for PPI in research is a dedicated PPI team, in place to raise awareness of, and provide expertise and support for PPI in research. This article reports on one of the UCLH BRC's major PPI initiatives—an extensive programme of training workshops in PPI for researchers. The programme of workshops is just one component of a continually evolving strategy for PPI at the BRC. It helps to illustrate some of the ways in which a large and complex research partnership can look to learn and innovate its research strategy. This article reports on how the UCLH BRC has developed and deployed an extensive training programme in PPI for research staff as a major component of embedding a more extensive culture of PPI in research.

## METHODS
### Training needs analysis
To inform the UCLH BRC's programme of PPI training, consultation was carried out with researchers and with UCLH patients.

### Needs analysis of researchers
Approximately 100 health researchers from UCL and UCLH were surveyed to ascertain education needs and preferences. Forty-eight per cent of respondents had not previously involved patients or the public in their research although 73% reported that they intended to do so. Respondents were asked to select their training preferences from a list of topics. Over 50% of respondents selected the topics of 'How to fill in the PPI section of a funding application', 'Taster/introduction to PPI', 'Practical guide to planning PPI' and 'Effective partnership working with charities' as their highest preferences. Less popular topics were 'Communicating biomedical research', 'Facilitation skills', 'Chairing meetings' and 'Setting up a patient advisory group'.

### Consulting with patients
The BRC has multiple patient panels and a network of patients who work with researchers. A facilitated discussion workshop with 12 people who had previously been actively involved in working with researchers in research design was carried out. People who could not attend were asked to complete a survey on what skills they felt researchers needed training in to carry out PPI. These exercises highlighted two main issues for patients. First, was a sense that researchers commonly needed support in improving how they communicate with patients, for instance, with more attention spent on enquiring and listening. The second was a view that researchers would benefit from greater understanding of the value of involving patients, particularly how patients could add value throughout a research project, providing practical help with the successful delivery of studies.

### Patient and public involvement
Training was developed and carried out in partnership with patients. Patients, who had experience of working with researchers as a part of PPI, worked with the trainers to identify and design the kind of training researchers would benefit from. This work informed the subject and format of the training workshops. It also informed the design of the surveys of workshop attendees, enabling us to focus on the issues and skills that patients had identified as a priority. A good example is researchers' communication skills, which patients had highlighted.

Workshops were delivered with a patient and a researcher and these cofacilitators continually fed back so that workshop design could be developed and improved.

### Training workshops
Drawing on the learning from the training needs analysis and discussions with patients, a programme of training workshops was established by the UCLH BRC team working closely with two very experienced, nationally recognised trainers with specific expertise and experience in PPI in research. The programme was designed to be appealing to biomedical researchers by focusing on practical tools for PPI that staff could deploy in their own

research. The programme was designed to be flexible and iterative, the trainers working with BRC staff and patients to continually adapt the content and format of the workshops to accommodate the needs of researchers that were identified during training.

The workshop topics were 'Introduction to PPI', 'How to fill in the PPI section of a grant/REC form', 'Accessing and sustaining patients and the public', Facilitating a group discussion with patients', 'PPI Masterclass', 'PPI in early-stage clinical laboratory research' and 'Effective partnership working in PPI'.

Workshops were advertised widely via the UCLH BRC website, and the organisational communication channels of UCLH, UCL and UCL partner organisations, as well as via the BRC's social media channels. Workshops were not targeted at specific staff groups or levels of seniority. For the first 3 years of the programme, each workshop was delivered by the same two people—both of whom were experienced trainers. One of them is also a patient advocate. They worked closely with a researcher and a patient who helped deliver each workshop and provide additional facilitation. In the final year of the programme (2018) reported in this article, the number of workshops was rationalised to three topics and delivered by one member of staff from the BRC PPI team. This was because a member of staff with training experience had been appointed and this enabled a greater number of workshops to be delivered over the academic year. The workshops typically had a half-day duration of between 3 and 4 hours. They were carried out at multiple different sites across UCL, UCLH and partner sites at Great Ormond Street, Moorfields Eye Hospital, UCL Partners and Queen Mary University London. No charge was made for attendance but a small non-attendance penalty fee was introduced after year 1 of the programme to discourage non-attendance of workshops that were often significantly over-subscribed.

### Workshop attendees
A total of 72 workshops were carried out over 5 years, 2014–2018. The workshops attracted 721 attendees from a variety of different professional groups and with a wide range of experience in biomedical research.

From 2018, attendees were awarded a UCL career point for every half-day workshop attended.

### Evaluation of the workshops
The evaluation was embedded into the design of the programme so the UCLH BRC could assess whether the workshops enabled attendees to translate their reflections and learning into research practice. Specifically,
1. Did training build up confidence and knowledge and enable researchers to carry out PPI they could not have done before?
2. What kind of PPI did they carry out and what effect did it have on research?

On arrival at the workshop, each participant was handed a survey to complete before the workshop (survey 1) and a sealed envelope containing a survey to complete

after the workshop (survey 2) before exiting the room. An average response rate of 98% was achieved for surveys 1 and 2. Six months after the workshops, a third survey (survey 3) was sent to all participants using an online tool and a response rate of 34% was achieved. Full results from survey 3 for the 2017/2018 attendees were not available for inclusion in this analysis. The survey questions can be viewed in Measures online supplemental file 1.

### Data analysis
All evaluation data were entered into Microsoft Excel where descriptive analysis was carried out.

## RESULTS
### Workshop attendees
The workshops were attended by staff with a wide range of research roles. Of the 721 attendees on the programme, data on the job roles is available on 649 (90%) staff. Thirty-one per cent of the attendees were medical consultants or scientists and 16% were research administrators and statisticians. Nearly half of the respondents listed their staff group as other, which included a wide range of roles such as dieticians, dentists, public health specialists, physiotherapists, clinical service managers and psychologists.

### Satisfaction levels and immediate impact of the workshops on researchers
Over 95% of participants each year rated workshops very good or excellent. Workshop attendees reported marked increases in the level of awareness of the resources available to help them with PPI after attending the training. Just 17% felt they knew about the resources prior to training, rising to 80% following training. Marked increases in levels of self-reported understanding of PPI were also reported after attending training rising from 27% to 86% after training (figure 1).

### Researchers' confidence and capabilities to do PPI
Both the first and second surveys asked about attitudes, understanding and competencies in PPI, to see whether training brought about any changes. Researchers reported increased confidence and capabilities in several areas of PPI following training (figure 2). Marked increases were found in self-reported levels of confidence to do PPI, run effective meetings and to involve patients and the public in steering groups.

### Impacts of the training 6 months later
The response rate for the 6 months survey was only 34% (2014–2017 182/540). However, the attendees who did respond provided useful insights into the longer-term impact of the training.

Six months after their workshop 65% of the responding attendees reported that they had carried out PPI. The main areas researchers reported that patients had helped with their research in were: prioritising research topics (45%); designing a study protocol (43%) and writing patient information materials and consent forms (36%) (figure 3).

Knowledge and understanding

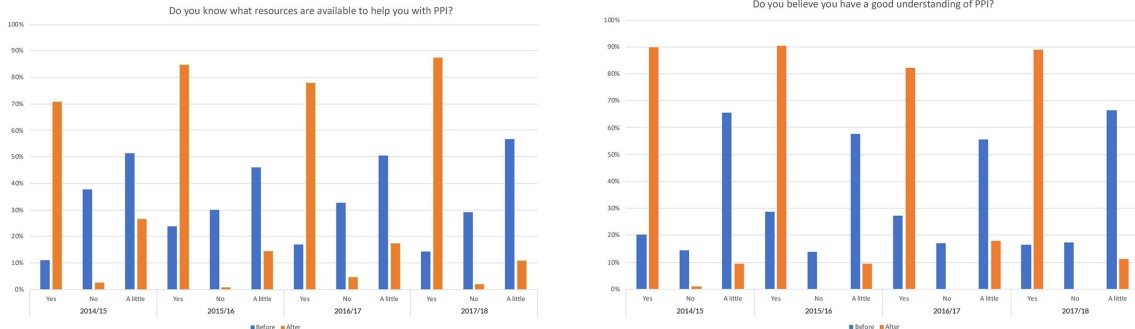

**Figure 1** Knowledge and understanding of PPI. PPI, patient and public involvement.

These findings were to some extent borne out in researchers' responses to questions about the main areas where the PPI had influenced their research. Over 30% had rewritten patient information materials and 36% had changed their study design. However, only 10% said their departmental/unit had made changes to research strategic direction and priorities (figure 4).

Approximately two-thirds of respondents at 6 months reported that they anticipated that PPI would ensure more meaningful outputs from their research. Nearly half predicted their research would have greater credibility with funders and stakeholders (figure 5).

Confidence to carry out PPI, was higher 6 months after training, rising from 58% just after training to 67% 6 months later.

## DISCUSSION

In summary, we have demonstrated that a large NIHR BRC can deliver an extensive training programme in PPI in research that caters for a wide variety of biomedical research professionals at a range of levels of seniority. This included research administrators who often get overlooked but play a vital role in research funding bids and set up of studies, as well as research delivery. The preparatory scoping activities we carried out, which involved researchers and patients, helped tailor the training. Moreover, the approach of iterating workshop content enabled us to respond to the needs of researchers, ensuring workshops were relevant to the research community. The programme strategically focused on practical skills for PPI to enable researchers to build their confidence in doing PPI while progressively acquiring the skills to put PPI into practice in their own research, from priority setting to co-delivery of research. This focus on enabling and encouraging researchers to carry out PPI is a different emphasis to studies that recommend work that places conceptual work including power as central.[23–25] An important baseline finding from our work was that, prior to the training, only 20% of attendees felt they were aware of the resources available to them to support PPI

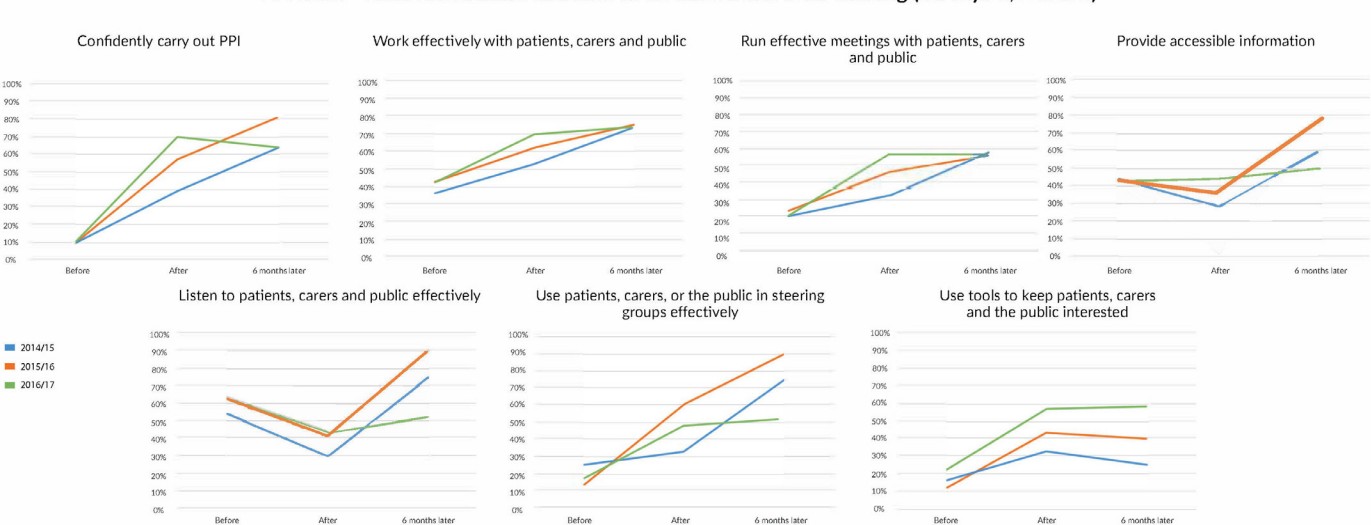

**Figure 2** PPI skills. PPI, patient and public involvement.

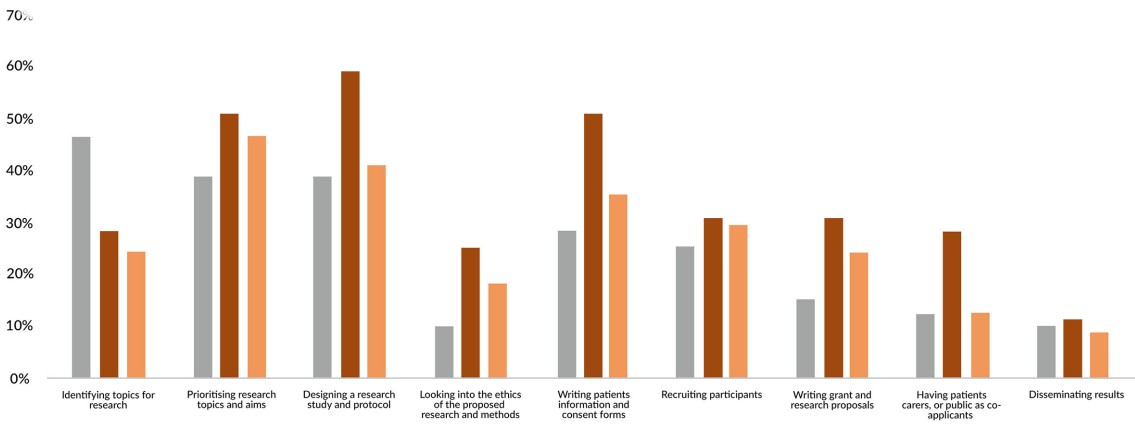

**Figure 3**  Kinds of PPI carried out. PPI, patient and public involvement.

in research. This finding, coupled with the fact that the training was generally well received by attendees, leads us to conclude that the workshops appeared to work well for people who had awareness that they required more skills to do PPI. Similar findings have been found in other studies of PPI in research training.[18] We are less able to draw conclusions about the benefits of the programme for people who already have higher level of experience and knowledge of available resources to do PPI. Although we did introduce an experience-level distinction in the workshops in 2015 by pitching some at 'beginners' and some at 'intermediate' level, this innovation was dropped in 2018 because we found attendees tended to ignore the distinction and attendees at all workshops were of mixed experience.

Further thought will be needed about how a training programme such as this can continue to evolve, building

in more content and experiential learning for research staff who have greater experience of doing PPI. PPI is inherently relational. As such, the best way to learn is to 'learn by doing' and to put into practice the practical skills acquired through the type of training that the UCLH BRC has implemented. Despite the limited response rate, the results of survey 3 would suggest first-hand experience of carrying out PPI after training helps to further increase researchers' confidence to carry out PPI. For the UCLH BRC, this is a journey. We plan to continually evolve our training, and involve our researchers and patients in the programme, re-engaging researchers for their own continued learning and enabling them to share their learning with other researchers. It will also become appropriate to review the purpose of training in PPI and consider whether it is primarily to encourage and enable researchers inexperienced in PPI to involve patients

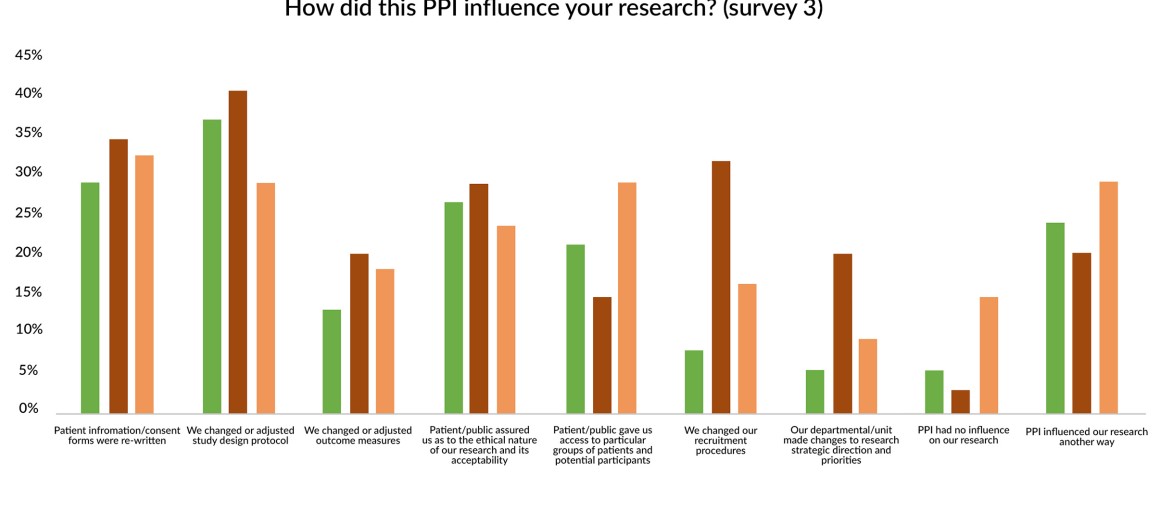

**Figure 4**  Impact of PPI on research. PPI, patient and public involvement.

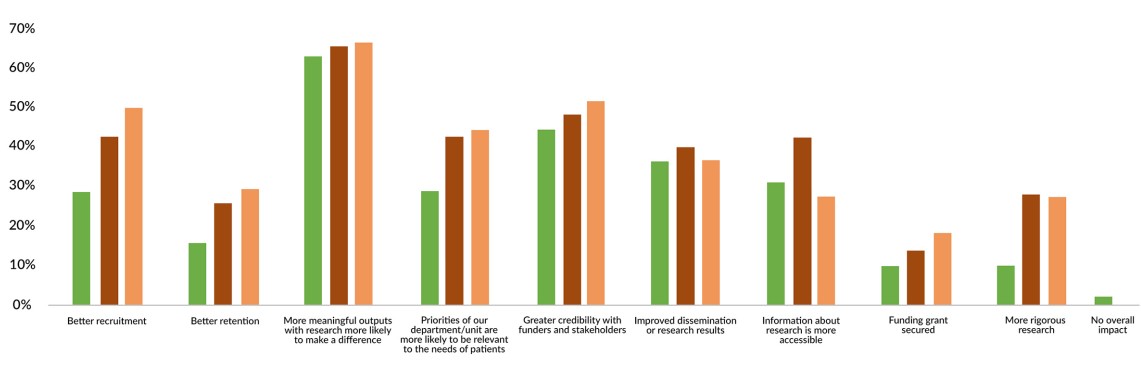

What do you anticipate will be the overall impact of this PPI on your research? (survey 3)

**Figure 5** Anticipated future impact of PPI on research. PPI, patient and public involvement.

and the public and discover the value of involvement, or whether there is also a need to support the more PPI-experienced researchers.

The high levels of self-reported improvements in understanding, knowledge and skills to do PPI are suggestive also of a strong benefit of the training. It is certainly true that evaluation responses focusing on levels of confidence and awareness to perform certain PPI activities recorded in the immediate aftermath of training are likely to be high as training content will still have been at the forefront of the attendee's minds at that juncture. Caution is needed not to over-interpret the findings from the 6 month survey given the low response rate and the fact that those who did respond at the 6 month stage were likely to have had more positive experience of doing, or planning for, PPI in their research in the 6 months following training. Nevertheless, there are interesting signals from the data that suggest increases in confidence and knowledge in PPI were carried forward into individuals' research activities. There were also signals that carrying out PPI after training was likely to build confidence and self-reported skills. The high level of consistency in the feedback from attendees in survey 2 (immediate) and survey 3 (6 months) across the 5 years of the programme adds further reassurance that the positive impacts of the training that we have observed are real.

Notwithstanding the limitations of the low response rate to the 6-month survey, the findings do suggest that, following attendance at the workshops, researchers doing PPI were more likely to pursue activities at the earlier stages of the research process, notably involving patients in prioritising research topics and designing studies. The lower levels of PPI at the stage of reporting and dissemination of results may be a reflection that taking PPI into practice takes time and researchers may prefer to take forward an incremental approach to doing PPI on their new studies. This would benefit from further exploration to fully understand whether the apparent front-loading of PPI activity in the research process is real and to

identify any training needs to support more PPI across the research lifecycle. The work also suggests several other areas that require further investigation, in particular the differences in impact of different kinds of involvement, and the extent to which experiential learning—learning by actually practising PPI—is key to developing Patient and Public Involvement and Engagement (PPIE) in biomedical research.

One significant drawback with our evaluation is that it is based on self-report by the researchers who attended the workshops. We have not yet sought to acquire the experiential feedback of the patients who have been involved in the research activity being carried out by researchers who attended the workshops. To fully understand the impact of the training for patients it will clearly be important to incorporate patients into evaluation given the relational underpinnings of good PPI.[20][21] It will also be important to build in considerations of the quality of PPI carried out, reflecting on the use of appropriate methods for PPI,[26] and on adherence to emerging standards.[2][20][21]

As more funding organisations demand PPI as part of the application process, the type of training that has been developed at the UCLH BRC will be very important. During the COVID-19 pandemic, the large-scale adoption of online tools has demonstrated the effectiveness of these media for meetings and training. Further development of the programme with online training options will provide the opportunity to involve more patients and incorporate other interactive approaches, such as quizzes, in the learning process. Attention also needs to be given to overcoming the problem of self-selection in PPI, and ensuring that there is an inclusive approach to involvement in research characterised by PPI being representative of population diversity.[27]

Central to the NIHR's standards for PPI is the need for researchers and research organisations to embed PPI into the culture of the organisation.[2] A training programme alone will not achieve that, particularly given the organisational complexity of major NHS–University partnerships

that have BRCs and the consequent power dynamics that prevail within these institutions.[12 28] At the UCLH BRC, the training programme sits within a wider context in which many other PPI and engagement activities are resourced and pursued. The UCLH BRC has a dedicated, experienced and accessible team of staff who coordinate our activities in PPI and engagement and provide support and mentorship for researchers. Each of the UCLH BRC's 11 scientific themes pursue theme-specific PPI activities.[29] We host a large annual Research Open Day in University College Hospital at which up to 50 research groups have displays to showcase their research for hundreds of patients and visitors to the hospital, encourage engagement with research and identify new opportunities for public involvement. We fund 50 laboratory placements with UCL biomedical scientists every year for school pupils from disadvantaged backgrounds.[30] We also work with researchers to develop innovative ways to communicate research with language and formats that are accessible. This package of broader support and resources helps build researchers' confidence and skill levels to do PPI alongside the BRC's structured training programme in PPI.

The ongoing challenge is to weave all of these complementary initiatives together to drive positive change, and high-quality PPI, in a large community of biomedical researchers. The extent to which these activities actually change researchers' practice is not easy to measure, especially in a large complex biomedical research partnership. The hope is that, within a broad and varied approach to PPI, such as that at the UCLH BRC, researchers will find things to inspire them to continue to explore good ways to involve patients in their research.

## KEY RECOMMENDATIONS

► Training of researchers should be considered a fundamental part of developing the involvement of the public in research and built into long-term strategic planning and investment.

► Training should be tailored to give researchers practical skills, building up their confidence, practical knowledge and the capacity to experience first-hand the value of PPI to research. This pragmatic approach can lay the foundations for the future by equipping a generation of researchers to involve patients and the public in research.

► Further evaluation is recommended to understand how beneficial to research PPI training is, or indeed whether patients perceive any impact of PPI on research.

**Acknowledgements** Special thanks to patient advisors Libby Cooper, Brenden Conroy, Philip Creasy, Anthony Locke.

**Contributors** RY (corresponding and lead author and guarantor) lead contribution: established and managed the training programme, designed the evaluation of the programme and carried out the analysis of findings, and wrote the first and part of the final draft of the article. SD equal contribution: leader of the work area and advised on production of the training, and its analysis, and edited and steered

the article from draft to final stage. BH equal contribution: one of the trainers commissioned, with another trainer, to design and carry out the training, including its evaluation, and edited and steered the article from draft to final stage. JA equal contribution: carried out analysis of statistical findings. NJM equal contribution: led and oversaw the Biomedical Research Centre team carrying out the training programme and managed its final outcome, and steered and edited the article from draft to final stage. Wrote the final draft of the manuscript.

**Funding** National Institute for Health Research Biomedical Research Centre at University College London Hospitals NHS Foundation Trust—award/grant number is not applicable. Health Education North Central and East London—grant WFDevProjects062. Wellcome—grant number 204841/Z/16/Z.

**Competing interests** All authors have completed the Unified Competing Interest form and, with two exceptions, declare: no support from any organisation for the submitted work; no financial relationships with any organisations that might have an interest in the submitted work in the previous 3 years, no other relationships or activities that could appear to have influenced the submitted work. One author discloses that they were a paid employee of the institution running the training discussed at the time this work was undertaken. Another author discloses that they received a fee to deliver the training discussed in the article.

**Patient and public involvement statement** Training was developed and carried out in partnership with patients. Patients, who had experience of working with researchers as a part of PPI, worked with the trainers to identify and design the kind of training researchers would benefit from. This work informed the subject and format of the training workshops. It also informed the design of the surveys of workshop attendees, enabling us to focus on the issues and skills that patients had identified as a priority. A good example is researchers' communication skills, which patients had highlighted. Workshops were delivered with a patient and a researcher and these co-facilitators continually fed back so that workshop design could be developed and improved.

**Patient consent for publication** Not required.

**Ethics approval** This work is an evaluation of a training programme and is not research. As such, Research Ethics Committee approval was not sought for the evaluation.

**Provenance and peer review** Not commissioned; externally peer reviewed.

**Data availability statement** Data are available upon reasonable request. Data are made up of answers to surveys by researchers attending training. Attendees are not identifiable. Data are not in a repository and can be requested from Rosamund.yu@ucl.ac.uk.

**ORCID iD**
Rosamund Yu http://orcid.org/0000-0002-6202-429X

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
