## [Reviewer comments · BMJ Open]

ARTICLE DETAILS

TITLE (PROVISIONAL)	Evaluation of a patient and public involvement training programme for researchers at a large biomedical research centre in the United Kingdom.
AUTHORS	Yu, Rosamund; Hanley, Bec; Denegri, Simon; Ahmed, Jaber; McNally, Nicholas

VERSION 1 – REVIEW

REVIEWER	Ní Shé, Éidín University of New South Wales, School of Population Health
REVIEW RETURNED	13-Feb-2021

GENERAL COMMENTS	This paper sets out to evaluate the impact of training in patient and public involvement on biomedical researchers and their research. On my reading of this paper, I asked myself, so what-i don't think the authors have provided much depth to this question nor has any been linking to the broader literature -Ref end after line 23 and no link to the evidence is made in the discussion-this needs to be improved. I suggest a major revision of this paper with some suggestions: 1. Introduction-rewrite needed to update the literature -e.g. 2011 study ref-now ten years old indeed things have moved on as funders have shifted-also line 25 makes a statement on charity sector with no evidence.2. Methods-surely without understanding the philosophical underpinning and values of PPI you are providing training to maintain tokenistic PPI (e.g. consent form development) and perhaps why many did not undertake reciprocal PPI Was the training an issue? -Would like to understand this more as for me PPI is about values.3. Profile of those who undertook the training-more detailed needed what level they were and the training itself.4. Did you map pre and post what type of PPI was being undertaken-what level, was it after the evaluation, e.g. were PPI members involved at the pre-commencement stage of idea development?5. The whole discussion needs to be rewritten a bit too much full of praise without much critiques or reflection from the literature of any consideration of the broader contextual challenges of developing long term PPI.
--

REVIEWER	Murphy, Edel NUI Galway, School of Medicine
REVIEW RETURNED	17-Mar-2021

GENERAL COMMENTS	This paper reports on an evaluation of a PPI workshop delivered
---

over four years to more than 700 scientists, clinicians and managers in a biomedical research centre. Significant time and resources have been and continue to be invested in introductory PPI training for researchers and clinicians and it is important to understand the impact of this training on research and research results. Therefore, this paper has the potential to be an important addition to the PPI literature. However some structural and methodological issues deserve more attention and, if addressed, will make a worthy and stronger contribution to the literature, for example:

- A clearer definition of the research question or objective. It is not clear what the evaluation aimed to find out and the evaluation questions are not provided, therefore it is not possible to assess whether the conclusions drawn are credible. If the workshops were "designed to build practical skills and confidence to work with patients and the public" (line 8 pg 3 of 16) the authors should consider whether this should be the focus of the evaluation. Was the evaluation based on any evaluation framework?

- A clearer more detailed Methods section would be very helpful, with more detail on both the workshop(s) and the evaluation process, for example:
Is this "a programme of PPI training workshops" or one workshop given repeatedly to different cohorts of attendees? A clearer description of the workshop(s) would be very helpful: outline content, length of workshop, and the format (eg was there opportunity to practice any new skills?). A clear statement that each attendee attended one such workshop (or not?) would be useful (if that is a correct interpretation). Inclusion of the evaluation questions would be very helpful.

- Greater consistency, particularly in the language used, in reporting the results would be beneficial. It is not clear why results are presented separately for the first year of the training and then presented for subsequent years - why not present all years as one? If there is a good reason for separating out the first year's data, please clarify. A clearer definition of what the aim of the evaluation (first point above) should provide guidance on how to structure the presentation of the results.

- Is it reasonable to attribute the level and nature of PPI activities after 6 months to attendance at one (or more?) workshops? Additionally, putting skills learned into practice, ("doing" PPI), has a recognised impact on the individual's confidence, using positively. Therefore, should the discussion consider if any factors other than the workshop(s) may be influencing the PPI activities?

- The argument and supporting data referenced around "the braver approach to involvement" (pg 10 of 16) are, as currently presented, confusing and therefore, unconvincing.

- The language used in the section Patient and Public Involvement (pg 5 of 16) is not convincing that the activities described were all in fact PPI. This is of particular concern in a paper about PPI training. For example "surveys" are usually considered a data-gathering research activity, not PPI. "Focus groups" is an ambiguous term, if the focus groups were underpinned by PPI values and conducted with a partnership rather than with a "data collection" focus perhaps "discussions" is a better word (cf Doria et al 2018 Res Invol Engagem 2018; 4:19). The involvement of a patient advocate in

	delivering the training is a clear example of PPI. - The language used does not always do justice to the work involved in gathering and analysing the data and at times makes it difficult to follow the narrative. For example: Line 59 pg 4 of 16, "emphasis on trying to 'get people through the door' seems at odds with the aim of developing high quality, effective PPI (line 5 pg 3 of 16) line 41 pg 5 of 16 "session were formatted".. Lines 45 to 49 pg 8 of 16, "baseline confidence.... sometimes double or go up by a third..." line 22 (pg 10 of 16) "something of a guaranteed impact" line 14 (pg 11 of 16) "lecture on the moral and ethical imperatives" - is this linked to the statement (pg 5 of 16) rather than on its philosophical underpinnings? - A statement on why this data collection and analysis did not require ethical approval and/or evidence of consultation with a REC on the need for ethics approval would be appropriate.
--	---

VERSION 1 – AUTHOR RESPONSE

The authors thank the reviewers for all their helpful feedback and comments. The comments have enabled the authors to revisit the article and restructure it to better contextualise the work and findings provide a clearer description of the methods and results sections. In fully taking on board the reviewers' comments, the authors feel that the paper provides a clearer articulation of the contribution this work makes to the important area of PPI training in a major NHS and University partnership.

We highlight below how the manuscript has been re-written:

Comment: This paper sets out to evaluate the impact of training in patient and public involvement on biomedical researchers and their research. On my reading of this paper, I asked myself, so what-i don't think the authors have provided much depth to this question nor has any been linking to the broader literature -Ref end after line 23 and no link to the evidence is made in the discussion-this needs to be improved.

Response: We thank the reviewer for this comment. The paper has now been comprehensively re-written. It now includes a better structured and more up to date discussion of the literature in the Introduction and more depth to the discussion at the end of the paper that is contextualised with more references to the available literature. Very few PPI training programmes in a biomedical research setting have been published so we feel that our paper offers an important contribution within this gap.

Comment: 1. Introduction-rewrite needed to update the literature -e.g. 2011 study ref-now ten years old indeed things have moved on as funders have shifted-also line 25 makes a statement on charity sector with no evidence.

Response: The Introduction has been comprehensively re-written and the references significantly updated to better set the context for our work.

Comment: 2. Methods-surely without understanding the philosophical underpinning and values of PPI you are providing training to maintain tokenistic PPI (e.g. consent form development) and perhaps why many did not undertake reciprocal PPI Was the training an issue? -Would like to understand this more as for me PPI is about values.

Response: More description of the content of the training programme has been provided in the

revised manuscript. The programme included a workshop on Background to PPI which itself included some content on the philosophical underpinnings of PPI and all workshops included exercises that questioned attitudes to PPI. This training programme was very specifically targeted at practical tools for researchers in PPI. This emphasis was informed by the training needs analysis we did with patients and with researchers as we were designing the training programme. The new manuscript includes some more details on the training needs analysis. We have also explained in the Discussion section that the training programme does not sit in isolation. It is part of a wider approach to PPI which, when taken together, are aimed at changing the culture for PPI at our institution.

Comment: 3. Profile of those who undertook the training-more detailed needed what level they were and the training itself.

Response We have included details of the professional groupings of the individuals who undertook the training in the revised manuscript.

Comment: 4. Did you map pre and post what type of PPI was being undertaken-what level, was it after the evaluation, e.g. were PPI members involved at the pre-commencement stage of idea development?

Response: The surveys did ask about PPI carried out before training by researchers. However, the data revealed no significant patterns and therefore the authors did not refer to this part of the evaluation.

Comment: 5. The whole discussion needs to be rewritten a bit too much full of praise without much critiques or reflection from the literature of any consideration of the broader contextual challenges of developing long term PPI.

Response: The whole paper has been rewritten including a comprehensive overhaul of the Discussion section. The revised version is a more balanced discussion of the pros and cons of our work with references to the literature where appropriate and discussion of the broader contextual challenges of developing PPI in the long term. We thank the reviewer for these comments. We agree that our work could make a stronger contribution to the literature in this really important area, which is characterised by very few published evaluations of PPI training programmes. We have therefore welcomed the opportunity to comprehensively revise our paper to give more attention to the key issues

Comment: This paper reports on an evaluation of a PPI workshop delivered over four years to more than 700 scientists, clinicians and managers in a biomedical research centre. Significant time and resources have been and continue to be invested in introductory PPI training for researchers and clinicians and it is important to understand the impact of this training on research and research results. Therefore, this paper has the potential to be an important addition to the PPI literature. However some structural and methodological issues deserve more attention and, if addressed, will make a worthy and stronger contribution to the literature, for example:

Response The paper reports on a PPI training programme for biomedical researchers and specifically on the key findings from an evaluation of the training. It was not, therefore, a research project but the evaluation did follow a structured approach. The revised manuscript provides more details about the evaluation questions and methodology we used. We have replied to the individual examples below

Comment: - A clearer more detailed Methods section would be very helpful, with more detail on both the workshop(s) and the evaluation process, for example:

Is this "a programme of PPI training workshops" or one workshop given repeatedly to different cohorts of attendees? A clearer description of the workshop(s) would be very helpful: outline content, length of workshop, and the format (eg was there opportunity to practice any new skills?). A clear statement that each attendee attended one such workshop (or not?) would be useful (if that is a correct interpretation). Inclusion of the evaluation questions would be very helpful.

Response: The Methods section of the paper has been comprehensively re-written to attend to these points. It now provides more description of what we did, the attendees and the format and content of the workshops themselves.

Comment: - Greater consistency, particularly in the language used, in reporting the results would be beneficial. It is not clear why results are presented separately for the first year of the training and then presented for subsequent years - why not present all years as one? If there is a good reason for separating out the first year's data, please clarify. A clearer definition of what the aim of the evaluation (first point above) should provide guidance on how to structure the presentation of the results.

Response: The paper has been comprehensively re-written to ensure consistency throughout the manuscript. The revised text covers the main observations from the figures and reference to the findings between years is included in the discussion as a way of drawing reassurance of the voracity of the observations. The aim of the evaluation is now more clearly stated in the revised manuscript.

Comment: - Is it reasonable to attribute the level and nature of PPI activities after 6 months to attendance at one (or more?) workshops? Additionally, putting skills learned into practice, ("doing" PPI), has a recognised impact on the individual's confidence, using positively. Therefore, should the discussion consider if any factors other than the workshop(s) may be influencing the PPI activities?

Response: We completely agree with this statement. We have addressed this point by including a more comprehensive discussion of the pros and cons of our work in the Discussion and by contextualising the training within our broader organisational PPI strategy which will also be driving changes in behaviour towards PPI and the culture for PPI.

Comment: - The argument and supporting data referenced around "the braver approach to involvement" (pg 10 of 16) are, as currently presented, confusing and therefore, unconvincing.

Response: This language has been removed from the revised manuscript

Comment: - The language used in the section Patient and Public Involvement (pg 5 of 16) is not convincing that the activities described were all in fact PPI. This is of particular concern in a paper about PPI training. For example "surveys" are usually considered a data-gathering research activity, not PPI. "Focus groups" is an ambiguous term, if the focus groups were underpinned by PPI values and conducted with a partnership rather than with a "data collection" focus perhaps "discussions" is a better word (cf Doria et al 2018 Res Involv Engagem 2018; 4:19). The involvement of a patient advocate in delivering the training is a clear example of PPI.

Response: The language in the Methods and Results sections has been improved to be more representative of the activities that were undertaken in the workshops. Reference is now made more broadly to 'Discussions' to reflect how these workshops were pitched.

We have included the Doria et al reference in the Discussion as part of a discussion about the quality of PPI.

We have also emphasised in the text the involvement of a patient advocate in the training programme.

Comment: The language used does not always do justice to the work involved in gathering and analysing the data and at times makes it difficult to follow the narrative. For example:

Line 59 pg 4 of 16, "emphasis on trying to 'get people through the door' seems at odds with the aim of developing high quality, effective PPI (line 5 pg 3 of 16) line 41 pg 5 of 16 "session were formatted"...

Lines 45 to 49 pg 8 of 16, "baseline confidence.... sometimes double or go up by a third..."

line 22 (pg 10 of 16) "something of a guaranteed impact"

line 14 (pg 11 of 16) "lecture on the moral and ethical imperatives" - is this linked to the statement (pg 5 of 16) rather than on its philosophical underpinnings?

Response;

The paper has been comprehensively re-written to address all issues of language and to ensure

consistency of language throughout the paper.

Comment: - A statement on why this data collection and analysis did not require ethical approval and/or evidence of consultation with a REC on the need for ethics approval would be appropriate.

Response: Clarifying statements are now included in the manuscript.

VERSION 2 – REVIEW

REVIEWER	Ní Shé, Éidín University of New South Wales, School of Population Health
REVIEW RETURNED	03-May-2021
GENERAL COMMENTS	None.
REVIEWER	Murphy, Edel NUI Galway, School of Medicine
REVIEW RETURNED	14-May-2021
GENERAL COMMENTS	Thank you for the excellent work done in revising this paper. It was a pleasure to read this clear, well-structured revision, which will now make an excellent contribution to the PPI literature.